# Unveiling Species Diversity Within Early-Diverging Fungi from China IX: Four New Species of *Mucor* (Mucoromycota)

**DOI:** 10.3390/jof11090682

**Published:** 2025-09-19

**Authors:** Zi-Ying Ding, Xin-Yu Ji, Fei Li, Wen-Xiu Liu, Shi Wang, Heng Zhao, Xiao-Yong Liu

**Affiliations:** 1College of Life Sciences, Shandong Normal University, Jinan 250300, China; 15270343451@163.com (Z.-Y.D.); ji15965902393@163.com (X.-Y.J.); lifeisdnu@126.com (F.L.); 18054356852@163.com (W.-X.L.); wangssdau@126.com (S.W.); 2CAS Key Laboratory of Forest Ecology and Silviculture, Institute of Applied Ecology, Chinese Academy of Sciences, Shenyang 110016, China; zhaoheng181@mails.ucas.ac.cn; 3Institute of Microbiology, Chinese Academy of Sciences, Beijing 100101, China

**Keywords:** *Mucorales*, basal fungi, fungal diversity, taxonomy, molecular phylogeny

## Abstract

*Mucor* species are fast-growing filamentous fungi, widespread in natural ecosystems. As opportunistic pathogens, some species can cause mucormycoses in humans and animals, while others hold significant economic value in food fermentation and bioengineering. In this study, four novel species were identified from soil samples collected in Xizang Autonomous Region and Yunnan Province, China, and their establishment as new species was supported by morphological characteristics and molecular data (ITS-LSU-*RPB1*), with phylogenetic analyses conducted using the Maximum Likelihood (ML) and Bayesian Inference (BI) methods. *M. globosporus* sp. nov. is characterized by producing globose chlamydospores. *M. multimorphus* sp. nov. is distinguished by swelling in the sporangiophores. *M. polymorphus* sp. nov. is differentiated by polymorphic chlamydospores. And *M. xizangensis* sp. nov. reflects its geographical origin in the Xizang Autonomous Region. Comprehensive descriptions of each novel taxon are presented herein. This study constitutes the ninth segment in an ongoing series elucidating early-diverging fungal diversity in China, expanding the understanding of the phylogeny of *Mucor* fungi and extending the worldwide number of known *Mucor* species to 137.

## 1. Introduction

The genus *Mucor*, a group of fast-growing, early-diverging fungi, is species-rich and distributed widely in natural ecosystems [1,2]. It is commonly found in soil, air, herbivore dung, insects, necromass of animals and plants, and other damp environments [3,4,5,6]. As coprophilic fungi, some species serve as pioneer decomposers during fecal decomposition, significantly contributing to material cycling in ecosystems through mediating the biogeochemical cycles of carbon and nitrogen [7,8]. Some species are opportunistic pathogens of animals, causing cutaneous mucormycoses in humans, especially in immunocompromised individuals [9,10,11]. Several *Mucor* species can also induce the spoilage of natural and artificial foods [3]. However, certain *Mucor* species exhibit significant application value in food industries, frequently used for fermenting soybean products, cheeses, and other foods [12,13]. And certain species play an important role in bioengineering by producing various enzymes such as lipases, proteases, phytases, cellulases, and uricases, which are vital for biocatalytic processes and industrial applications [14,15,16,17,18,19,20].

The genus was originally described by Fresenius in 1850 [21] and characterized by simple or branched sporangiophores arising directly from substrates, non-apophysate and globose sporangia with persistent or deliquescent, incrusted sporangial walls, and zygospores borne on opposed suspensors [22,23]. However, the morphological classification of *Mucor* remains contentious. For example, the traditional taxonomic literature historically differentiated *Rhizomucor* from *Mucor* based on rhizoids [24,25], but recent molecular studies demonstrate that certain *Mucor* species also produce rhizoids (e.g., *Mucor changshaensis*) [5,26,27]. This explains why some *Mucor* species were misclassified into the genus *Rhizomucor*. Most *Mucor* species are mesophilic, exhibiting optimal growth at 20–30 °C and survival within 10–42 °C [28]. By contrast, a minority are psychrophilic, characterized by an optimal growth temperature of approximately 15 °C, a minimum growth temperature capable of reaching 0 °C, and a maximum growth temperature in the vicinity of 20 °C [29,30].

Since its formal inclusion in Linnaeus’ *Species Plantarum* (1753) [31], the genus *Mucor* has witnessed continuous taxonomic refinements, from the initial morphological delineation to the contemporary phylogenetic reclassification. In 2018, Wijayawardene et al. stated that among more than 300 literature-recorded *Mucor* species, only approximately 60 species were valid or could be validated [32]. Recent taxonomic revisions integrating phylogenetic analyses and morphological characters have led to the systematic reclassification of multiple species previously assigned to *Mucor* into other genera [11]. Nevertheless, the discovery and formal description of novel species in recent years have further enriched the taxonomic diversity of the genus *Mucor* [5,27,33,34,35]. Currently, 133 species are accepted (https://www.catalogueoflife.org/, accessed on 30 June 2025).

During the investigation of soil fungal diversity in China, eight strains were classified into four new *Mucor* species based on ITS-LSU-*RPB1* molecular data, morphological characteristics, and maximum growth temperatures. Phylogenetic trees, detailed descriptions, and photographs of these new taxa are presented herein. This is the ninth report of a serial work on the diversity of early-diverging Chinese fungi [36,37,38,39,40,41,42,43].

## 2. Materials and Methods

### 2.1. Sample Collection and Strain Isolation

Soil samples were collected from Xizang Autonomous Region and Yunnan Province, China, between 2022 and 2024—with two sampling expeditions in Xizang Autonomous Region and eight in Yunnan Province. Sampling was carried out during the rainy season or the vigorous vegetation growth period in both regions. Yunnan sampling sites were located in subtropical evergreen broad-leaved forests, coniferous-broad-leaved mixed forests, and alpine meadows, with dominant red and yellow soils and high vegetation coverage. Xizang Autonomous Region sites focused on alpine meadows, shrubs, and sparse valley forests, with alpine meadow soils as the main soil type. All samples were collected from 5 to 10 cm depth, labeled with a waterproof tag, indicating the collection date, administrative division, GPS coordinates, and altitude. These samples were then temporarily stored in a 4 °C incubator in the laboratory.

Subsequently, strains were isolated following the plate dilution method and single spore isolation method described in previous studies [44,45]. In short, 1 g of soil sample was weighed and suspended in 10 mL of sterile water to prepare a soil suspension with a concentration of 10^−1^. Accurate 1 mL of this suspension was transferred to 9 mL sterile water and mixed thoroughly to obtain a soil suspension with a concentration of 10^−2^. Serial dilutions were continued to prepare soil suspensions at concentrations of 10^−3^ and 10^−4^. About 200 μL of 10^−3^ and 10^−4^ soil suspensions were drawn by a sterile pipette and spread evenly onto the surface of Rose Bengal Chloramphenicol agar [46] (RBC: peptone 5.00 g/L, glucose 10.00 g/L, MgSO_4_·7H_2_O 0.50 g/L, KH_2_PO_4_ 1.00 g/L rose bengal 0.05 g/L, chloramphenicol 0.10 g/L, agar 15.00 g/L) containing 0.03% streptomycin sulfate. The coated plates were incubated in the dark at 25 °C for 3–5 d. After sporangia produced, individual spores were picked under a stereomicroscope (Olympus SZX10, OLYMPUS, Tokyo, Japan) using a sterile inoculation loop and inoculated onto Potato Dextrose Agar (PDA: 200 g potato, 20 g dextrose, 20 g agar, 1000 mL distilled water, pH 7.0). The inoculated PDA plates were incubated at 25 °C in the dark. Strains were purified and stored with 10% glycerol at 4 °C.

The type strains were deposited at the China General Microbiological Culture Collection Center, Beijing, China (CGMCC) and duplicated at Shandong Normal University, Jinan, China (XG). The dried specimens were stored at the Herbarium Mycologicum Academiae Sinicae, Beijing, China (Fungarium, HMAS). Taxonomic information of these new taxa was submitted to the fungal name database (https://nmdc.cn/fungalnames/, accessed on 30 June 2025).

### 2.2. Morphology and Maximum Growth Temperature

Colonies were cultured on PDA medium at 25 °C for 2–5 days. Macro- and microscopic structures were observed daily. Tape-stripping and wet-mount methods were adopted in the slide preparation process.

A high-definition color digital camera (DP80, Olympus, Tokyo, Japan) was used to photograph both the obverse and reverse sides of the colonies. A stereomicroscope (Olympus SZX10, Olympus, Tokyo, Japan) and an optical microscope (BX53, Olympus, Tokyo, Japan) were used to observe microscopic structures (including hyphae, rhizoids, stolons, sporangiophores, sporangia, collars, columellae, apophyses, sporangiospores, chlamydospores, and zygospores). After that, Adobe Photoshop CC 2019 was used to layout of different microstructures images. Then, Digimizer software (https://www.digimizer.com/, accessed on 30 June 2025) was systematically utilized to measure various dimensions of the microstructures, with the statistical data covering 20 measurements.

The maximum growth temperature was performed following the methodology described by previous study [47,48,49]. To determine the maximum growth temperature of the target fungal strain, the strain was initially cultured at 25 °C for 3 d, followed by a controlled daily temperature increase of 1 °C until colony formation halted. Three independent parallel replicates were incorporated in the design for statistical reliability.

### 2.3. DNA Extraction, PCR Amplification, and Sequencing

After the target strains were incubated at 25 °C for 5–7 d on PDA solid medium plates, cell total DNAs were extracted from the mycelia using the BeaverBeads Plant DNA Kit [50] (Cat. No.: 70409–20; BEAVER Biomedical Engineering Co., Ltd., Suzhou, China). For strains with inefficient genomic DNA amplification, reverse transcription was performed to synthesize cDNA from total RNA as an alternative template using SPARKscript II All-in-one RT SuperMix for qPCR (With gDNA Eraser) (Cat# AG0305-B; Shandong Sparkjade Biotechnology Co., Ltd., Jinan, China). Genomic loci ITS, LSU, and *RPB1* were amplified by polymerase chain reaction (PCR) using the primer pairs ITS4/ITS5 [51], LR0R/LR7 [52], and RPB1-Af/RPB1-Cr [53], respectively (Table 1). PCR amplification was carried out using a 25μL reaction system, including 12.5μL of 2 × Hieff Canace^®^Plus PCR Master Mix (Yeasen Biotechnology, Shanghai, China, Cat No. 10154ES03), 10 µL ddH_2_O, 1 μL of each of the forward and reverse primers (10 µM) (TsingKe, Beijing, China), and 1 μL fungal genomic DNA template. The amplification products were detected by 1% agarose gel electrophoresis, and the band specificity was observed after staining with TS-GelRed Nucleic Acid Gel Stain (10,000 × in water; TSJ002; Beijing Tsingke Biotech Co., Ltd., Beijing, China). Gel recovery was carried out using a Gel Extraction Kit (Cat# AE0101-C; Shandong Sparkjade Biotechnology Co., Ltd., Jinan, China). DNA sequencing was performed by the Biosune Company Limited (Shanghai, China). Consensus sequences were obtained through MAFFT v.7.0 alignment, and assembled with MEGA v.7.0. Then all sequences were uploaded to the GenBank database, with the accession numbers provided in Appendix A.

### 2.4. Phylogenetic Analyses

The phylogenetic analyses were constructed based on concatenated ITS-LSU-*RPB1* sequences, using both the Maximum Likelihood (ML) and Bayesian Inference (BI) methods. The optimal evolutionary model of each locus was determined by MrModelTest v2.3 [54] and subsequently applied in the Bayesian inference (BI) analysis. The ML analysis was conducted using RAxML-HPC2 on XSEDE v.8.2.12 on the CIPRES Science Gateway platform (https://www.phylo.org/, accessed on 30 June 2025), with 1000 bootstrap replicates [55]. The BI analysis was conducted on a Linux system server, with a quick start configured with an automatic stop option. Bayesian inference consisted of five million generations with four parallel runs, employing stopping rules and sampling frequencies of 100 generations [56]. The burn-in score was set to 0.25, and the posterior probability (PP) was determined based on the remaining trees. The evolutionary trees were uploaded to the iTOL website (https://itol.embl.de, accessed on 30 June 2025) for layout and adjustment. The final refinements were carried out using Adobe Illustrator CC 2019.

## 3. Results

### 3.1. Phylogeny

The molecular dataset included 83 strains in total, consisting of 46 *Mucor* species, with *Backusella lamprospora* (CBS 118.08) as an outgroup. The dataset consisted of 3290 characters, covering ITS rDNA (1–1172), LSU rDNA (1173–2283), and *RPB1* (2284–3290). Among them, 1808 characters were constant, 246 variable characters were parsimony-uninformative, and 1236 characters were parsimony-informative. The results of the MrModelTest analysis indicated that the Dirichlet base frequencies and the GTR + I + G evolutionary model were suitable for the two partitions in the Bayesian inference. The topology of the Maximum Likelihood (ML) tree, consistent with that of the Bayesian Inference (BI) tree, was chosen as a representative for detailed illustration (Figure 1). Eight strains of *Mucor* isolated in this study were divided into four independent clades: *M. globosporus* (MLBV = 100, BIPP = 1.00), *M. multimorphus* (MLBV = 100, BIPP = 1.00), *M. polymorphus* (MLBV = 100, BIPP = 1.00), and *M. xizangensis* (MLBV = 100, BIPP = 1.00). As for phylogenetic relationship, *M. globosporus* is closely related with *M. moniliformis*, *M. multimorphus* and *M. xizangensis* are sisters to each other, and *M. polymorphus* is basal to *M. multimorphus* and *M. xizangensis*.

### 3.2. Taxonomy

#### 3.2.1. *Mucor globosporus* Z.Y. Ding, H. Zhao & X.Y. Liu, sp. Nov., Figure 2

Fungal Names—FN 573008.

**Figure 2 jof-11-00682-f002:**
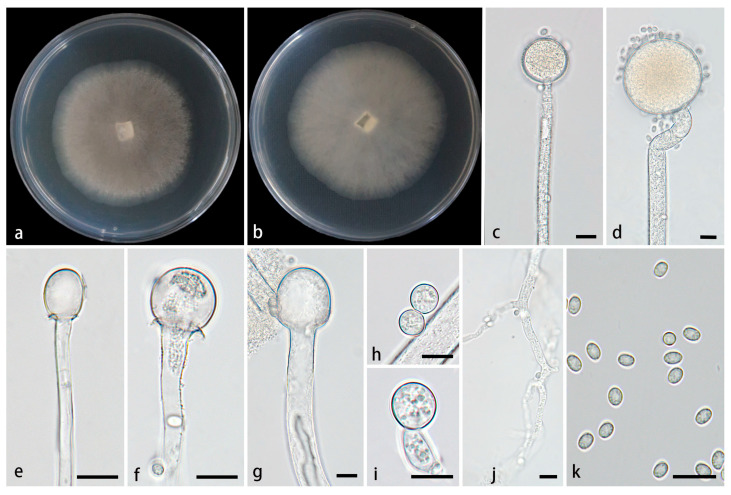
*Mucor globosporus* ex-holotype CGMCC 3.28970. (**a**,**b**) Colonies cultured on PDA at 25 °C for 3 days, (**a**) obverse, (**b**) reverse; (**c**,**d**) sporangia; (**e**–**g**) columellae; (**h**,**i**) chlamydospores; (**j**) rhizoids; (**k**) sporangiospores. Scale bars: (**c**–**k**) 10 μm.

Type—China, Yunnan Province, Xishuangbanna Dai Autonomous Prefecture, Menghai County, Mengman Town, G219 (West Scenic Line) (22°08′01″ N, 100°18′87″ E, altitude 1367.38 m), from soil, 6 July 2024, Z.Y. Ding, holotype HMAS 354077, ex-type living culture CGMCC 3.28970 (=XG09777-10-1).

Etymology—The epithet *globosporus* (Lat.) refers to globose chlamydospores in this species.

Description—Colonies on PDA at 25 °C for 3 d, reaching 59 mm in diameter, rapidly growing with a growth rate of 19.7 mm/d, initially white, gradually becoming black-brown, floccose. *Hyphae* flourishing, occasionally branched, hyaline, aseptate when young, septate with age, radial growth. *Rhizoids* present, but rare. *Stolons* absent. *Sporangiophores* arising from substrate and aerial hyphae, erect or few slightly bent, unbranched, hyaline, occasionally with a swelling, 4.8–14.3 µm wide. *Sporangia* globose, pale yellow to light brown, 16.6–76.1 μm in diameter. *Collars* present or absent, if present usually distinct and large. *Columellae* subglobose, globose, ellipsoidal, hyaline or subhyaline, smooth-walled, 5.2–30.2 µm long and 4.5–27 µm wide. *Apophyses* absent. *Sporangiospores* usually ovoid, rarely globose, 3.0–5.1 µm long and 2.6–3.7 µm wide. *Chlamydospores* rare, globose, 6.6–11.3 µm long and 6.6–11.2 µm wide. *Zygospores* unknown.

Cultured characteristics and maximum growth temperature: Under the same culture conditions, the colonies grow faster on PDA than on MEA (Figure 3). On PDA, the colonies reach 73 mm in diameter for 5 d at 25 °C. On MEA, the colonies reach 58 mm in diameter for 5 d at 25 °C. No growth was obversed at 33 °C.

Additional strains examined—China, Yunnan Province, Xishuangbanna Dai Autonomous Prefecture, Menghai County, Mengman Town (22°11′02″ N, 100°17′22″ E, altitude 1492.96 m), from soil, 6 July 2024, Z.Y. Ding, living culture XG09770-10-2.

GenBank accession numbers—CGMCC 3.28970 (ITS, PV819211; LSU, PV833754; *RPB1*, PX048331), XG09770-10-2 (ITS, PV819212; LSU, PV833755; *RPB1*, PX048332).

Notes—Based on the ITS-LSU-*RPB1* phylogenetic tree, two strains of the *Mucor globosporus* sp. nov. formed a fully supported lineage (MLBV = 100, BIPP = 1.00; Figure 1), which is sister to *M. moniliformis*. Morphologically, the new species is distinguished from *M. inflatus* in sporangia, columellae, chlamydospores and rhizoids. The new species produces subglobose, globose and ellipsoidal columellae, while *M. moniliformis* only has the first two shapes. The chlamydospores of the new species are only globose, while *M. moniliformis* exhibits various shapes including ellipsoidal, ovoid, subglobose, globose or irregular. Additionally, rhizoids are present in the new species, whereas they are absent in *M. moniliformis* [27].

#### 3.2.2. *Mucor multimorphus* Z.Y. Ding, H. Zhao & X.Y. Liu, sp. Nov., Figure 4

Fungal Names—FN 573010.

**Figure 4 jof-11-00682-f004:**
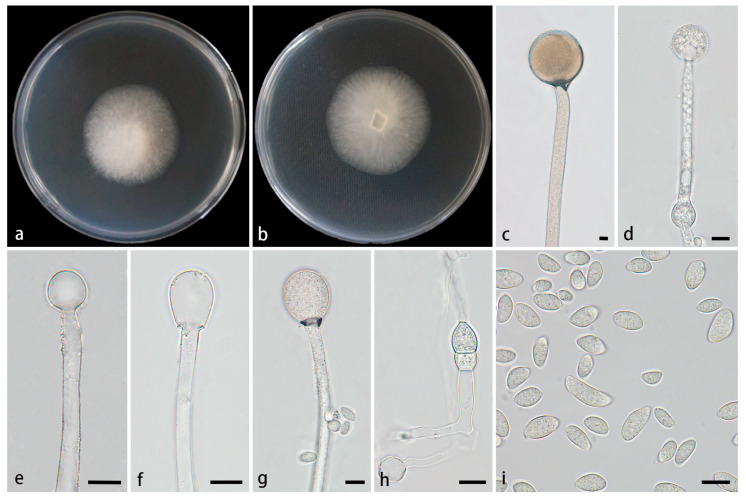
*Mucor multimorphus* ex-holotype CGMCC 3.28968. (**a**,**b**) Colonies cultured on PDA at 25 °C for 3 days, (**a**) obverse, (**b**) reverse; (**c**,**d**) sporangia; (**e**–**g**) columellae; (**h**) chlamydospores; (**i**) sporangiospores. Scale bars: (**c**–**i**) 10 μm.

Type—China, Xizang Autonomous Region, Xigaze City, Jilong County (The latitude and longitude are not clear, altitude 3040 m), from soil, 1 August 2022, Z.Y. Ding, holotype HMAS 354075, ex-type living culture CGMCC 3.28968 (=XG00398-8-1).

Etymology—The epithet *multimorphus* (Lat.) refers to producing multiple shapes of sporangiospores.

Description—Colonies on PDA at 25 °C for 9 d, reaching 77 mm in diameter, slowly growing with a growth rate of 8.56 mm/d, initially white, gradually becoming Cream yellow with age, floccose. *Hyphae* flourishing, usually unbranched, hyaline, occasionally septate, radial growth. *Rhizoids* absent. *Stolons* absent. *Sporangiophores* arising from substrate and aerial hyphae, erect or few slightly bent, unbranched, hyaline, sometimes accompanied by a swelling, 5.0–15.8 µm wide. *Septa* sometimes present in sporangiophores. *Sporangia* globose, pale yellow to pale brown, 33.8–70.0 μm in diameter. *Collars* present, usually small. *Columellae* globose, ellipsoidal, pyriform, hyaline or subhyaline, smooth-walled, 10.1–40.0 µm long and 7.5–39.7 µm wide. *Apophyses* absent. *Sporangiospores* multiple shaped, mainly fusiform and ellipsoidal, occasionally irregular, 5.4–18.0 µm long and 3.3–7.8 µm wide. *Chlamydospores* produced in substrate hyphae, ellipsoidal or irregular, occasionally present, 6.3–16.8 µm long and 8.1–12.5 µm wide. *Zygospores* unknown.

Cultured characteristics and maximum growth temperature: Under the same culture conditions, the colonies grow faster on PDA than on MEA (Figure 5). On PDA, the colonies reaching 55 mm in diameter for 5 d at 25 °C. On MEA, the colonies reaching 45 mm in diameter for 5 d at 25 °C. No growth was obversed at 31 °C.

Additional strains examined—China, Xizang Autonomous Region, Xigaze City, Jilong County (The latitude and longitude are not clear, altitude 3040 m), from a soil sample, 1 August 2022, Z.Y. Ding, living culture XG00398-8-2.

GenBank accession numbers—CGMCC 3.28968 (ITS, PV819207; LSU, PV833750; *RPB1*, PV889321), XG00398-8-2 (ITS, PV819208; LSU, PV833751; *RPB1*, PV889322).

Notes—In the phylogenetic tree of ITS-LSU-*RPB1*, two strains of the *Mucor multimorphus* sp. nov. formed a fully supported independent clade (MLBV = 100, BIPP = 1.00; Figure 1), which is closely related to *M. xizangensis*. Morphologically, the new species differs from *M. xizangensis* in sporangiophores, sporangia, sporangiospores, and chlamydospores. It occasionally forms a swelling on sporangiophores, while *M. xizangensis* does not. It is larger than *M. xizangensis* in sporangia (33.8–70 μm vs. 23.3–58.6 μm). In sporangiospores and chlamydospores, it differs from *M. xizangensis* by larger size and more shapes. Specifically, it produces predominantly fusiform and ellipsoidal sporangiospores (5.4–18.0 × 3.3–7.8 μm) and ellipsoidal or irregular chlamydospores (6.3–16.8 μm × 8.1–12.5 μm), whereas *M. xizangensis* forms mainly ellipsoidal sporangiospores (3.9–8.4 × 2.4–4.9 μm) and globose chlamydospores (4.0–11.1 μm × 3.9–10.7 μm). Physiologically, the maximum growth temperature of the new species is 1 °C lower than that of *M. xizangensis* (30 °C vs. 31 °C).

#### 3.2.3. *Mucor polymorphus* Z.Y. Ding, H. Zhao & X.Y. Liu, sp. Nov., Figure 6

Fungal Names—FN 573009.

**Figure 6 jof-11-00682-f006:**
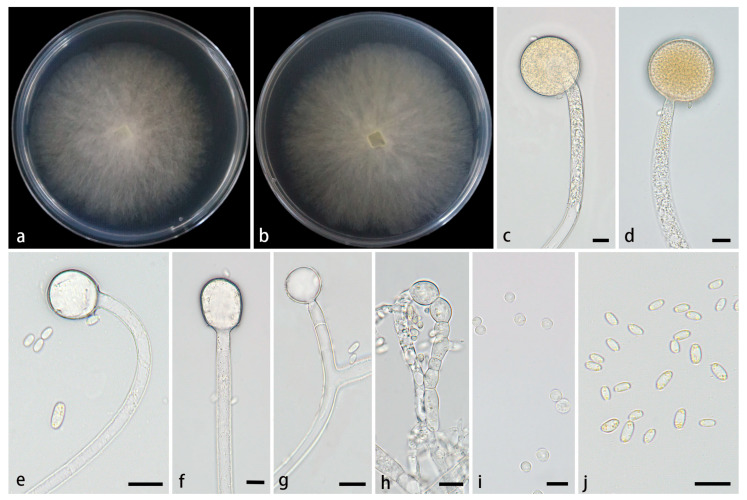
*Mucor polymorphus* ex-holotype CGMCC 3.28969. (**a**,**b**) Colonies cultured on PDA at 25 °C for 3 days, (**a**) obverse, (**b**) reverse; (**c**,**d**) sporangia; (**e**–**g**) columellae; (**h**,**i**) chlamydospores; (**j**) sporangiospores. Scale bars: (**c**–**j**) 10 μm.

Type—China, Yunnan Province, Puer City, Simao District, Yixiang Town, Yutang Section, Longlongba Jinyu Tea Estate (22°68′48″ N, 101°07′32″ E, altitude 1549.97 m), from soil, 5 July 2024, Z.Y. Ding, holotype HMAS 354076, ex-type living culture CGMCC 3.28969 (=XG09597-11-1).

Etymology—The epithet *polymorphus* (Lat.) refers to the polymorphic chlamydospores in this species.

Description—Colonies on PDA at 25 °C for 3 d, reaching 69 mm in diameter, rapidly growing with a growth rate of 23 mm/d, initially white, gradually becoming yellowish-brown with age, floccose. *Hyphae* flourishing, unbranched, hyaline, aseptate when juvenile, septate with age, radial growth. *Rhizoids* absent. *Stolons* absent. *Sporangiophores* arising from substrate and aerial hyphae, erect or few slightly bent, occasionally branched, hyaline, 2.2–14.4 µm wide. *Septa* sometimes present in sporangiophores. *Sporangia* globose, pale yellow to light brown, 36.7–49.0 μm in diameter. *Collars* present or absent, usually small. *Columellae* globose, ovoid, ellipsoidal, hyaline or subhyaline, smooth-walled, 5.1–28.3 µm long and 5.8–24.1 µm wide. *Apophyses* absent. *Sporangiospores* usually fusiform, 3.7–7.0 µm long and 1.9–3.6 µm wide. *Chlamydospores* produced in substrate hyphae, in chains, globose, ovoid, cylindrical or irregular, 4.5–17.2 µm long and 3.9–13.9 µm wide. *Zygospores* unknown.

Cultured characteristics and maximum growth temperature: Under the same culture conditions, the colonies grow faster on PDA than on MEA (Figure 7). On PDA, the colonies reaching 82 mm in diameter for 5 d at 25 °C. On MEA, the colonies reaching 80 mm in diameter for 5 d at 25 °C. No growth was obversed at 32 °C.

Additional strains examined—China, Yunnan Province, Puer City, Simao District, Yixiang Town, Yutang Section, Longlongba Jinyu Tea Estate (22°68′48″ N, 101°07′32″ E, altitude 1549.97 m), from soil, 5 July 2024, Z.Y. Ding living culture XG09597-11-2.

GenBank accession numbers—CGMCC 3.28969 (ITS, PV819209; LSU, PV833752; *RPB1*, PV948857), XG09597-11-2 (ITS, PV819210; LSU, PV833753; *RPB1*, PV948858).

Notes—Based on the ITS-LSU-*RPB1* phylogenetic tree, two strains of the *Mucor polymorphus* sp. nov. formed a fully supported clade (MLBV = 100, BIPP = 1.00; Figure 1), which is closely related to *M. multimorphus* and *M. xizangensis*. Morphologically, the new species is distinguished from *M. multimorphus* and *M. xizangensis* by sporangiophores, columellae, sporangiospores, and chlamydospores. Compared with *M. multimorphus*, the new species exhibits thinner sporangiophores (2.2–14.4 μm vs. 5.0–15.8 μm), smaller columellae (5.1–28.3 × 5.8–24.1 μm vs. 10.1–40.0 × 7.5–39.7 μm), and smaller sporangiospores (5.1–28.3 × 1.9–3.6 μm vs. 5.4–18.0 × 3.3–7.8 μm). Moreover, the new species produces globose, ovoid, and ellipsoidal columellae and fusiform sporangiospores, while *M. multimorphus* forms globose, ellipsoidal, and pear-shaped columellae and fusiform or ellipsoidal sporangiospores. Regarding chlamydospores, the new species produces chain-like with globose, ovoid, cylindrical or irregular ones, while *M. multimorphus* forms ellipsoidal or irregular ones. In contrast to *M. xizangensis*, the new species features thinner sporangiophores (2.2–14.4 μm vs. 5.0–17.8 μm), smaller columellae (5.1–28.3 × 5.8–24.1 μm vs. 7.9–30.1 × 7.7–29.4 μm), smaller sporangiospores (5.1–28.3 × 1.9–3.6 μm vs. 3.9–8.4 × 2.4–4.9 μm). And *M. xizangensis* produces globose, ellipsoidal, ovoid, and pyriform columellae, ellipsoidal sporangiospores and globose chlamydospores. Physiologically, the maximum growth temperature of the new species is 1 °C higher than those of *M. multimorphus* and the same as those of *M. xizangensis*.

#### 3.2.4. *Mucor xizangensis* Z.Y. Ding, H. Zhao & X.Y. Liu, sp. Nov., Figure 8

Fungal Names—FN 572032.

**Figure 8 jof-11-00682-f008:**
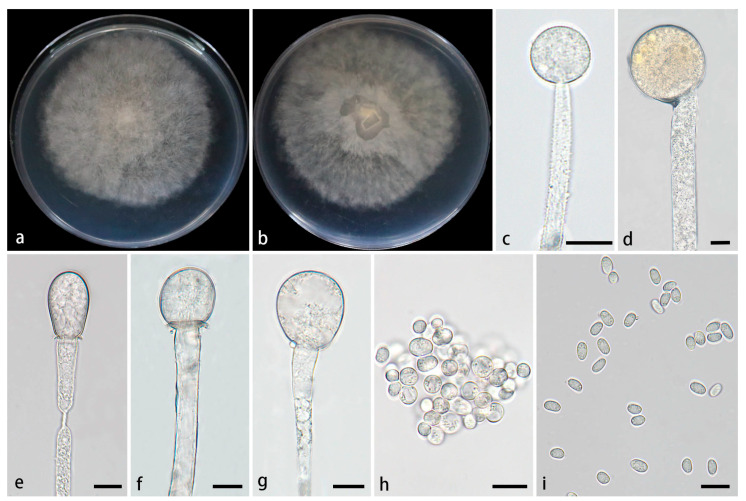
*Mucor xizangensis* ex-holotype CGMCC 3.28971. (**a**,**b**) Colonies cultured on PDA at 25 °C for 5 days, (**a**) obverse, (**b**) reverse; (**c**,**d**) sporangia; (**e**–**g**) columellae; (**h**) chlamydospores; (**i**) sporangiospores. Scale bars: (**c**–**i**) 10 μm.

Type—China, Xizang Autonomous Region, Nyingchi City and Milin City, close to the Yarlung Zangbo Grand Canyon (29°64′36″ N, 94°88′53″ E, altitude 2779.72 m), from soil, 29 August 2024, X.Y. Liu, holotype HMAS 354078, ex-type living culture CGMCC 3.28971 (=XG10424-13-1).

Etymology—The epithet *xizangensis* (Lat.) refers to the location, Xizang Autonomous Region, China, where the ex-holotype was collected.

Description—Colonies on PDA at 25 °C for 5 d, reaching 80 mm in diameter, rapidly growing with a growth rate of 16 mm/d, initially white, gradually becoming grayish-white, floccose. *Hyphae* flourishing, unbranched, hyaline, radial growth. *Rhizoids* absent. *Stolons* absent. *Sporangiophores* arising from substrate and aerial hyphae, erect or few slightly bent, unbranched, hyaline, 5.0–17.8 µm wide. *Sporangia* globose, white to light grayish-brown, 23.3–58.6 μm in diameter. *Collars* present or absent, if present usually distinct and large. *Columellae* globose, ellipsoidal, ovoid, sometimes pyriform, hyaline or subhyaline, smooth-walled, 7.9–30.1 µm long and 7.7–29.4 µm wide. *Apophyses* absent. *Sporangiospores* usually ellipsoidal, 3.9–8.4 µm long and 2.4–4.9 µm wide. *Chlamydospores* produced in substrate hyphae, mainly globose, 4.0–11.1 µm long and 3.9–10.7 µm wide. *Zygospores* unknown.

Cultured characteristics and maximum growth temperature: Under the same culture conditions, the colonies grow faster on PDA than on MEA (Figure 9). On PDA, the colonies reaching 80 mm in diameter for 5 d at 25 °C. On MEA, the colonies reaching 75 mm in diameter for 5 d at 25 °C. No growth was obversed at 32 °C.

Additional strains examined—China, Xizang Autonomous Region, Nyingchi City and Milin City, close to the Yarlung Zangbo Grand Canyon (29°64′36″ N, 94°88′53″ E, altitude 2779.72 m), from soil, 29 August 2024, Z.Y. Ding, living culture XG10424-13-2.

GenBank accession numbers—CGMCC 3.28971 (ITS, PV819213; LSU, PV833756; *RPB1*, PV973985), XG10424-13-2 (ITS, PV819214; LSU, PV833757; *RPB1*, PV973986).

Notes—Based on the ITS-LSU-*RPB1* phylogenetic tree, two strains of the *Mucor xizangensis* sp. nov. formed a fully supported independent lineage (MLBV = 100, BIPP = 1.00; Figure 1), which is closely related to *M. multimorphus*. Morphologically, the new species differs from *M. multimorphus* in sporangiophores, sporangia, sporangiospores, and chlamydospores. Specifically, the species does not form swelling on sporangiophores, while *M. multimorphus* occasionally develops a swelling on these structures. The sporangia of the new species are smaller than those of *M. multimorphus* (23.3–58.6 μm vs. 33.8–70.0 μm). In sporangiospores and chlamydospores, the new species is distinguished from *M. multimorphus* by smaller size and fewer shapes. More precisely, the new species produces predominantly ellipsoidal sporangiospores (3.9–8.4 × 2.4–4.9 μm) and globose chlamydospores (4.0–11.1 μm × 3.9–10.7 μm), while *M. multimorphus* develops fusiform and ellipsoidal sporangiospores (5.4–18.0 × 3.3–7.8 μm) and ellipsoidal or irregular chlamydospores (6.3–16.8 μm × 8.1–12.5 μm).

## 4. Morphological Comparison and Key to the Genus *Mucor* in China

Together with the four new *Mucor* species proposed, 28 species of this genus have been found in China. Since the detailed description of *Mucor gigasporus* was not available, the remaining 27 species were compared (Appendix A), and a concise key to these species was provided. The characteristics used in the key include colonies, hyphae, sporangiophores, sporangia, collars, columellae, sporangiospores, chlamydospores, and zygospores. Zygospores known………………………………………………..……….*M. homothallicus*Zygospores unknown…………………………………………………………………….....2Aborted sporangia known……………………………………………………………….....3Aborted sporangia unknown……………………………………………………………....6Fertile sporangia globose only………………………………………………….……..…...4Fertile sporangia globose and subglobose………………………………...……………...5Fertile sporangia 19.0–30.0 μm diameter……………………………*M. abortisporangium*Fertile sporangia 19.0–34.5 μm diameter………………………………………*M. radiates*Columellae 7.5–24.0 μm diameter…………………………………..…*M. chlamydosporus*Columellae 17.0–49.5 μm diameter……………………………….…….…… *M. orientalis*Collars known………………………………………………………………………………..7Collars unknown…………………………………………………………..……………….22Hyphae branched………………………………………………………….……..………….8Hyphae unbranched……………………………………………...…….…….………..…..19Sporangiospores shape > 2 kinds…………………………………………………..……...9Sporangiospores shape ≤ 2 kinds……………………………………………....………...10Fertile sporangia 17.0–52.5 μm diameter…………………………….……*M. breviphorus*Fertile sporangia 33.8–70.0 μm diameter…………………………….…*M. multimorphus*Sporangiophores branched………………………………………….…………….…..….11Sporangiophores unbranched………………………………………………………...….14Sporangia > 65 μm diameter……………………………………………………………...12Sporangia < 65 μm diameter……………………………………………………………...13Sporangiospores 4.0–7.0 × 3.5–6.0 μm………………………………………...*M. robustus*Sporangiospores 4.5–10.5 × 3.0–7.0μm…………………………….…..*M. sinosaturninus*Sporangiospores 4.0–7.0 × 3.0–5.0 μm…………………………………..*M. changshaensis*Sporangiospores 3.5–5.5 × 2.5–3.5 μm……………………….…………...*M. moniliformis*Chlamydospores known……………………………………………………………….….15Chlamydospores unknown……………………………………………………………….18Sporangia > 38 μm diameter………………………………………………………...……16Sporangia < 38 μm diameter……………………………………………………...………17Columellaesubglobose, globose, ellipsoidal……………………………...*M. globosporus*Columellae hemispherical or depressed globose………………….....*M. hemisphaericus*Sporangiospores usually ellipsoid, occasionally reniform………..*M. heilongjiangensis*Sporangiospores fusiform or ellipsoid……………………………….…....*M. rhizosporus*Collars obvious………………………………...……………………………*M. amphisporus*Collars not obvious………………………….…………………….…….*M. fusiformisporus*Chlamydospores known……………………………………………………………….....20Chlamydospores unknown……………………………………………………………….21Sporangiospores usually fusiform……………………………….……….*M. polymorphus*Sporangiospores usually ellipsoidal……………………………………….*M. xizangensis*Columellae multiple shaped, with pyriform……………………….…………….*M. tofus*Columellae multiple shaped, without pyriform…………………..………*M. brunneolus*Chlamydospores known…………………………………………………………….…….23Chlamydospores unknown……………………………………………………………….26Fertile sporangia > 44 μm diameter………………………………………………..…….24Fertile sporangia < 44 μm diameter…………………………..…………........................25Columellae elongated-conical, cylindrical……………………………..*M. chuxiongensis*Columellae globose to subglobose………………………………………*M. donglingensis*Columellae hemispherical or depressed globose…………………………...*M. floccosus*Columellae conical to cylindrical………………………………………..*M. hyalinosporus*Sporangiophores unbranched……………………………………………….…..*M. lobatus*Sporangiophores branched……………………………………….…………..…..*M. rongii*

## 5. Discussion

The first description of the genus *Mucor* dates back to 1850 [21], and since then its taxonomic studies have been continuously deepened. In 2016, Spatafora et al. [2] conducted phylogenetic analyses based on genomic data, segregating the *Mucor*-lineage fungi from the traditional phylum Zygomycota and establishing a distinct phylum, *Mucoromycota*. This taxonomic revision precisely defined their phylogenetic position in the fungal kingdom.

Modern fungal taxonomy relies predominantly on molecular data as the primary criterion to establish new taxonomic groups or evaluate interspecific relationships. Classical morphological characteristics (e.g., hyphal structure and spore morphology) and physiological traits (e.g., temperature tolerance) remain essential supplements for species delimitation. In phylogenetic research, integrating multigene markers such as ITS, LSU [25,26,57], and protein-coding genes like *RPB1* is crucial for resolving the evolutionary relationships in taxonomically complex lineages [28]. Most *Mucor* species delimitation studies commonly use ITS and LSU as genetic markers due to their high availability, while genes like *RPB1* aid in fine-scale analyses. Phylogenetic inferences via Maximum Likelihood (ML) and Bayesian Inference (BI) consistently showed that the novel species occupy stable phylogenetic positions with strong statistical support.

In this study, four novel *Mucor* species (*M. globosporus* sp. nov., *M. multimorphus* sp. nov., *M. polymorphus* sp. nov., and *M. xizangensis* sp. nov.) from China were identified through the integration of molecular data of ITS, LSU, and *RPB1*, combined with phenotypic observation and physiological trait assessments. Additionally, a systematic comparison of the morphological characteristics between these four novel species and their close relatives was performed (Table 2).

*Mucor globosporus* is closely related to *M. moniliformis.* In contrast to *M. moniliformis*, *M. globosporus* possesses distinct columellae and chlamydospore shapes, and rhizoids. *Mucor multimorphus* and *M. xizangensis* are sister taxa. Morphologically, *M. multimorphus* exhibits larger sporangia, and its sporangiospores and chlamydospores are both larger in size and more varied in shape. Additionally, *M. multimorphus* occasionally forms swellings on sporangiophores. *Mucor polymorphus* is closely related to *M. multimorphus* and *M. xizangensis.* Compared with the latter two, *M. polymorphus* has thinner sporangiophores, smaller columellae, and smaller sporangiospores. Moreover, the shapes of its columellae, sporangiospores, and chlamydospores differ significantly.

Physiologically, the thermal tolerance thresholds of these novel taxonomic groups were determined using a temperature gradient cultivation technique. Growth characterization revealed significant differences in the maximum growth temperatures among the four new *Mucor* species: *M. globosporus* 32 °C, *M. multimorphus* 30 °C, *M. polymorphus* 31 °C, and *M. xizangensis* 31 °C. These temperature parameters are consistent with the mesophilic physiological characteristics of most *Mucor* species, further supporting the taxonomic affiliation of these novel species groups with the *Mucor* genus.

Over the past five years, at least 45 new species of the genus *Mucor* have been discovered and described from diverse habitats such as soil, insects, plants, fungi, Mao-tofu, and animal dung (http://www.indexfungorum.org/, accessed on 30 June 2025), suggesting that this taxon retains extensive distribution potential in numerous understudied habitats. The four new species described herein further update the globally recognized species count of the genus to 137.

As a region of high fungal diversity, China contributes substantially to global *Mucor* diversity. Based on existing records, the inclusion of these four new species brings the total number of recognized *Mucor* species in China to 28 (https://www.catalogueoflife.org/, accessed on 30 June 2025), accounting for approximately 20% of the globally recognized *Mucor* species pool. This high regional richness is presumably attributed to the diverse climate zones and ecosystems in China. Taxonomic sampling for fungi in China has primarily focused on provinces with distinct ecosystems such as Yunnan, Sichuan, Guangdong, and Fujian, where intensive surveys have also facilitated *Mucor* discoveries. Notably, Yunnan leads in the description of new fungal species, including *Mucor* species. Beyond *Mucor*, China’s fungal diversity has also helped refine global fungal diversity estimates by filling gaps in East Asian mycobiota data, improving global distribution models, and expanding knowledge of extremophilic soil fungi.

Despite progress in characterizing *Mucor*’s species diversity and global distribution, its ecological functions have not been systematically analyzed, and *Mucor* groups in some extreme habitats remain underinvestigated. Future research should therefore prioritize under-sampled regions. Based on the above, this study not only provides new materials for *Mucor* taxonomy but also further highlights the necessity of continuously exploring their diversity and ecological functions.

## Figures and Tables

**Figure 1 jof-11-00682-f001:**
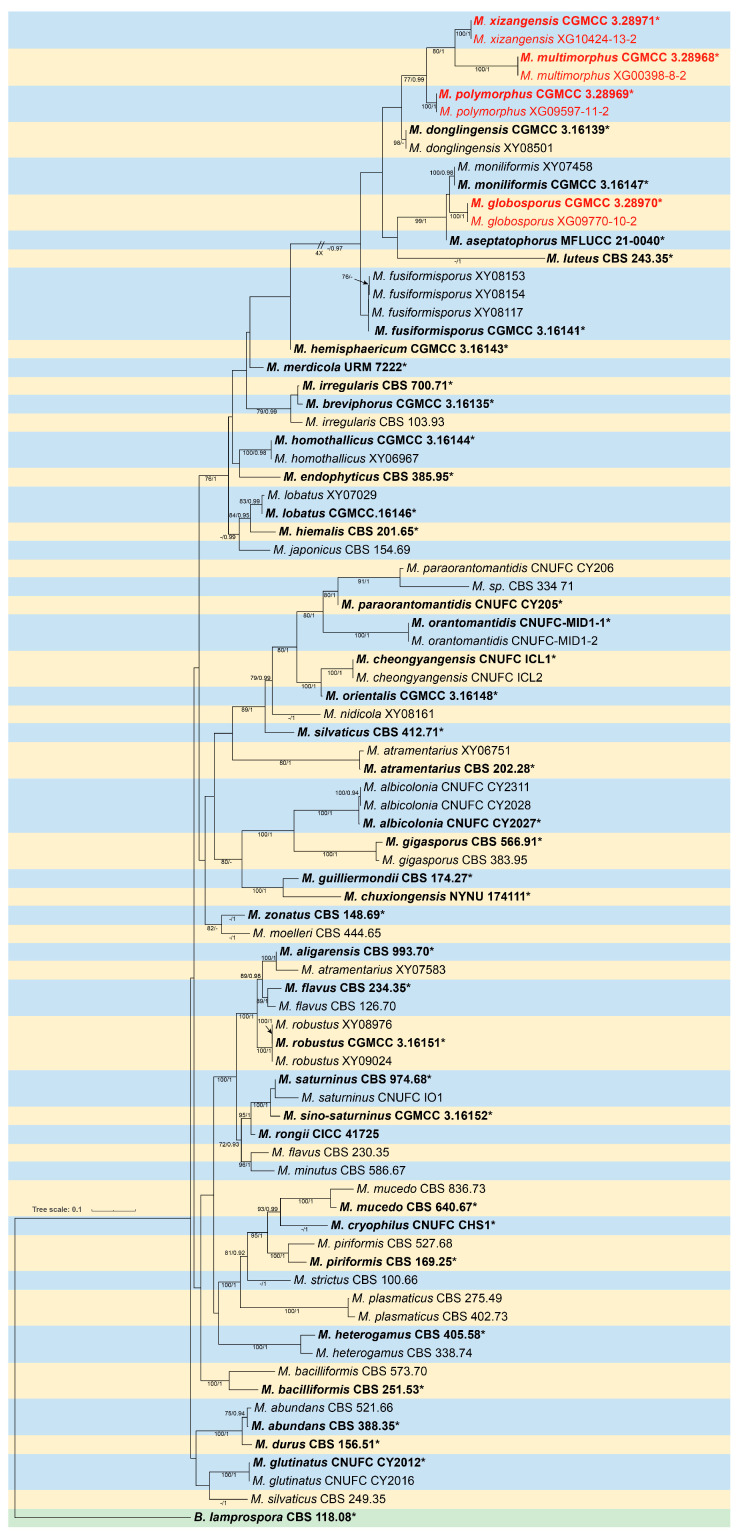
Maximum Likelihood (ML) phylogenetic tree of *Mucor* based on ITS, LSU, and *RPB1* sequences, with *Backusella lamprospora* (CBS 118.08) as outgroup. Nodes are labeled with ML bootstrap values (MLBV ≥ 70%) and Bayesian inference posterior probabilities (BIPP ≥ 0.9), separated by a slash “/”. Ex-type or ex-holotype strains are indicated in bold black and marked with an asterisk “*”. Strains isolated in this study are shown in red. The scale bar in the lower left represents 0.1 substitutions per site.

**Figure 3 jof-11-00682-f003:**
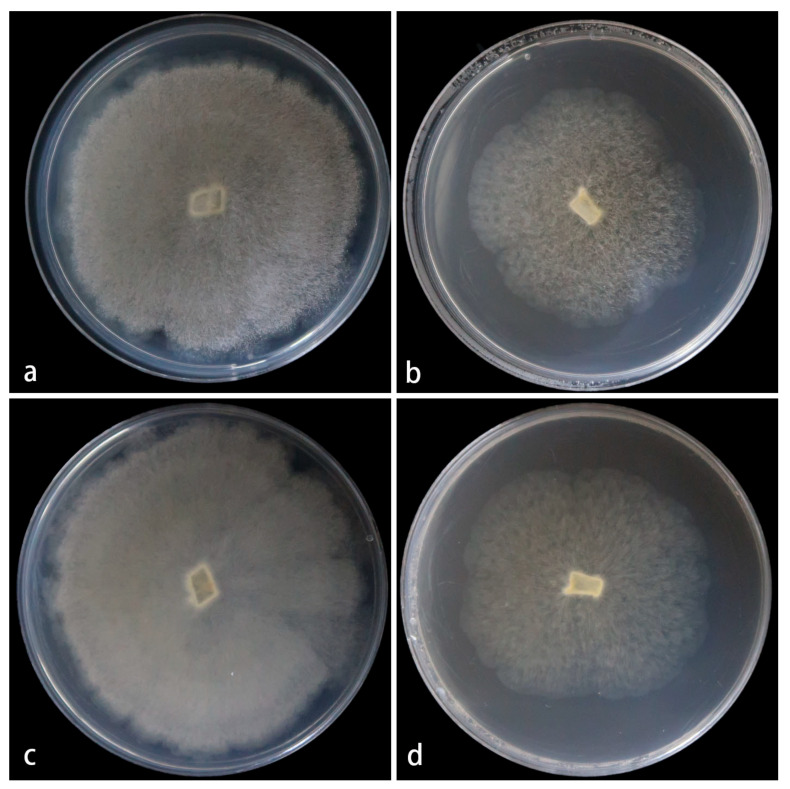
*Mucor globosporus* ex-holotype CGMCC 3.28970. Colonies cultured on PDA and on MEA at 25 °C for 5 days, (**a**,**c**) Colony on PDA; (**b**,**d**) Colony on MEA.

**Figure 5 jof-11-00682-f005:**
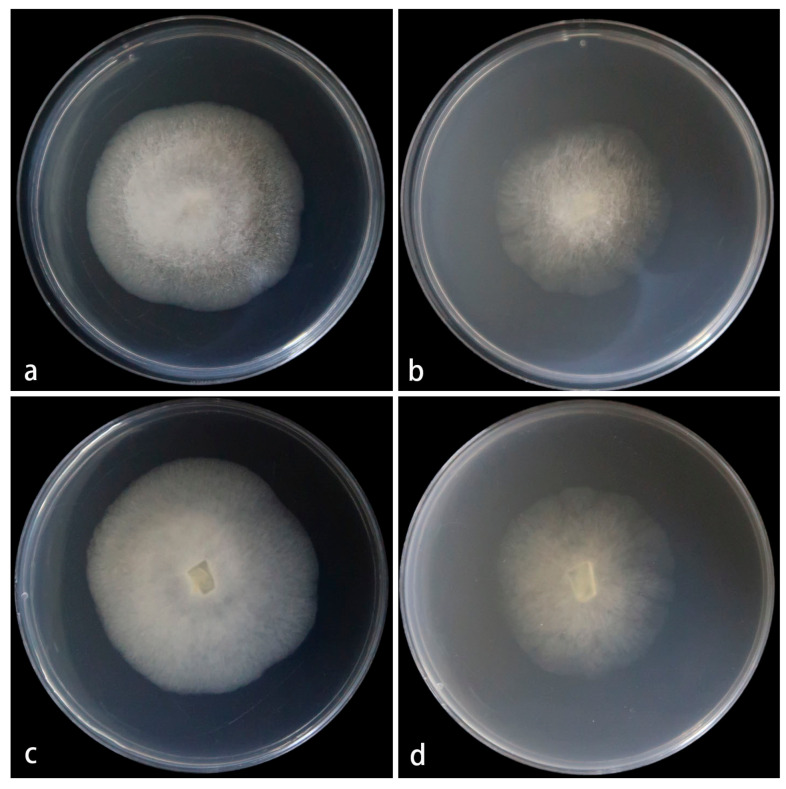
*Mucor multimorphus* ex-holotype CGMCC 3.28968. Colonies cultured on PDA and on MEA at 25 °C for 5 days, (**a**,**c**) Colony on PDA; (**b**,**d**) Colony on MEA.

**Figure 7 jof-11-00682-f007:**
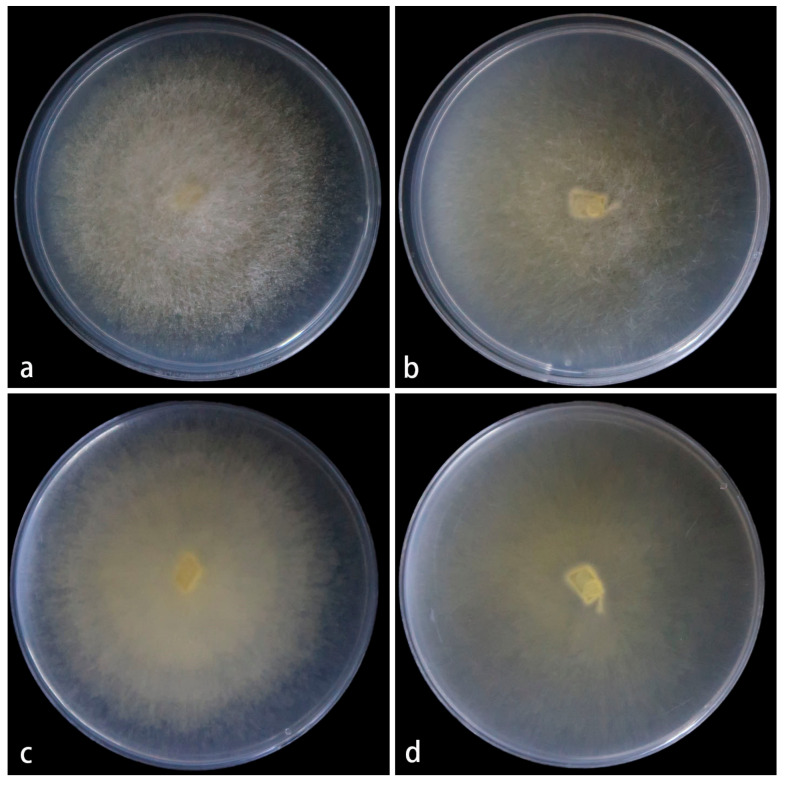
*Mucor polymorphus* ex-holotype CGMCC 3.28969. Colonies cultured on PDA and on MEA at 25 °C for 5 days, (**a**,**c**) Colony on PDA; (**b**,**d**) Colony on MEA.

**Figure 9 jof-11-00682-f009:**
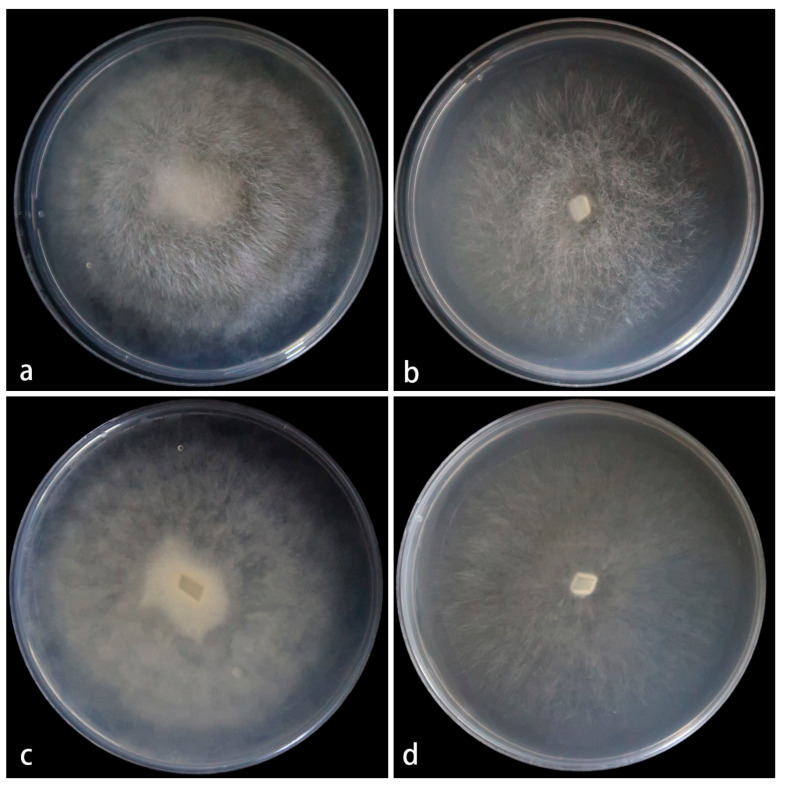
*Mucor xizangensis* ex-holotype CGMCC 3.28971. Colonies cultured on PDA and on MEA at 25 °C for 5 days, (**a**,**c**) Colony on PDA; (**b**,**d**) Colony on MEA.

**Table 1 jof-11-00682-t001:** PCR primers and programs used in this study.

Loci	PCR Primers	Primer Sequence (5′–3′)	PCR Cycles	References
ITS	ITS5 ITS4	GGA AGT AAA AGT CGT AAC AAG G TCC TCC GCT TAT TGA TAT GC	95 °C 5 min; (95 °C 30 s, 55 °C 30 s, 72 °C 1 min) × 35 cycles; 72 °C 10 min	[51]
LSU	LR0R LR7	GTA CCC GCT GAA CTT AAG C TAC TAC CAC CAA GAT CT	95 °C 5 min; (95 °C 50 s, 47 °C 30 s, 72 °C 1.5 min) × 35 cycles; 72 °C 10 min	[52]
*RPB1*	RPB-Af RPB-Cr	GAR TGY CCD GGD CAY TTY GG CCN GCD ATN TCR TTR TCC ATR TA	95 °C 3 min; (94 °C 40 s, 60 °C 40 s, 72 °C 2 min) × 37 cycles; 72 °C 10 min	[53]

**Table 2 jof-11-00682-t002:** Morphological characteristics of *Mucor* species involved in this study.

Species	Colonies	Sporangiophores	Sporangia	Columellae	Sporangiospores	Chlamydospores	Reference
*M. globosporus*	PDA: 25 °C 3 d, 59 mm, 19.7 mm/d, initially white, gradually becoming black-brown, floccose	unbranched, hyaline, 4.8–14.3 µm wide	globose, pale yellow to light brown, 16.6–76.1 μm diam.	1.2–5.8 µm long	mostly spherical, 7.3–14 × 7.8–13.4µm, with or without spines, 0.7–1.9 µm long	globose, 6.6–11.3× 6.6–11.2 µm	This study
*M. multimorphus*	PDA: 25 °C 9 d, 77 mm, 8.56 mm/d, initially white, gradually becoming cream yellow, floccose	unbranched, 5.0–15.8 µm wide	globose, pale yellow to pale brown, 33.8–70.0 μm diam.	globose, ellipsoidal, pyriform, 10.1–40.0 × 7.5–39.7 µm	mainly fusiform and ellipsoidal, occasionally irregular, 5.4–18.0 × 3.3–7.8 µm	ellipsoidal or irregular, 6.3–16.8 × 8.1–12.5 µm	This study
*M. polymorphus*	PDA: 25 °C 3 d, 69 mm, 23 mm/d, initially white, gradually becoming yellowish-brown, floccose	occasionally branched, 2.2–14.4 µm wide	globose, pale yellow to light brown, 36.7–49.0 μm diam.	globose, ovoid, ellipsoidal, smooth-walled, 5.1–28.3 × 5.8–24.1 µm	fusiform, 3.7–7.0 × 1.9–3.6 µm	in chains, globose, ovoid, cylindrical or irregular, 4.5–17.2 × 3.9–13.9 µm	This study
*M. xizangensis*	PDA: 25 °C 5 d, 80 mm, 16 mm/d, initially white, gradually becoming grayish-white, floccose	unbranched, 5.0–17.8 µm wide	globose, white to light grayish-brown, 23.3–58.6 μm diam.	globose, ellipsoidal, ovoid, pyriform, 7.9–30.1 × 7.7–29.4 µm	usually ellipsoidal, 3.9–8.4 × 2.4–4.9 µm	mainly globose, 4.0–11.1 × 3.9–10.7 µm	This study
*M. moniliformis*	PDA: 27 °C 5 d, 90 mm, 10 mm high, floccose, granulate, initially white soon becoming buff-yellow, reverse irregular at margin	simply branched, somewhere slightly constricted, 6.0–13.5 μm wide	globose, light brown or dark brown when old, 18.0–64.5 μm diam.	subglobose and globose, 15.0–30.5 × 17.0–32.0 μm	ovoid, hyaline or subhyaline, 3.5–5.5 × 2.5–3.5 μm	in chains, ellipsoidal, ovoid, subglobose, globose or irregular, 5.5–17.0 × 7.0–15.5 μm	[27]

## Data Availability

The original contributions presented in this study are included in the article/Appendix A. Further inquiries can be directed to the corresponding author.

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
