# Peer review of "Unveiling Species Diversity Within Early-Diverging Fungi from China IX: Four New Species of Mucor (Mucoromycota)"

_jof, 2025, doi:10.3390/jof11090682_

Round 1
Reviewer 1 Report
Dear Editor and authors,
I am very grateful for the opportunity to review this manuscript. It is good to know that there are many species of mucoraceous fungi being discovered in China. The paper is very interesting, as it describes four new species of Mucor isolated from soil based on morphology, phylogenies (three markers), and maximum growth temperature. The article is well written. I made several comments throughout the text on points that need correction, including the species descriptions, as many numbers need to be rounded.
Below are comments on the main corrections that need to be made, while other corrections are highlighted in the attached file.
1 - Line 16: …M. inflatus 15 sp. nov. distinguished by swelling in the sporangiophores.
Because swellings are uncommon, this cannot be used as the main distinguishing feature of this species. The specific epithet of this species must be modified.
2 - Line 17: …and M. xizangensis sp. nov. featuring its collection in Xizang…
What is the main morphological characteristic of this species? Authors must say that.
3 - Line 44: ….zygospores borne on opposed or apposed suspensors [22, 23].
As far as I know, suspensors in Mucor are always opposed, never apposed. Please check this.
4 - Line 70: 2.1. Sample collection and strain isolation
I really think you must provide additional information regarding your collection areas, as this information can be very useful for future ecology/biogeography studies. In which biomes are the collection areas located? How different are these areas to each other in terms of temperature, rainfall, vegetation and soil structure? Another important information is when the soil collections were carried out. How many soil sampling expeditions were carried out in the areas? Was the soil sampling the same for all collection areas? Was isolation done in triplicate?
4 - Line 93: Morphology and maximum growth temperature
This item needs to be improved by adding information about the morphological study. For example, which temperature was used to describe the specimens? What culture media have been used? How were the slides mounted for observation? How many measurements were made for each fungal structure? Please provide information on that.
5 - Line 101: The maximum growth temperature
Author must grow strains in at least two culture media in order to access the radial growth and maximum temperature growth.
6 - In the leads of figures 2 onwards: Make it clear how old the cultures are and what the incubation temperature is.
7 - Lines 395-399: This paragraph sounds like results to me, not discussion. I suggest the authors to discuss the using of maximum growth temperature as an additional tool for delimiting Mucor species, and even within the order Mucorales.
8 - Due to the large number of Mucor species described from China, I would like to see in this paper an identification key for the species of these genera described from this country.

Reviewer 2 Report
Dear Editor,
I reviewed this manuscript and found that it is interesting, but needs a revision.
In this manuscript, authors have introduced four new species of Mucor from China.
All the necessary changes have mentioned in the relevant places.
In some places, needs to rewrite the sentences/words or check the grammer.
Ovarall, I would like to see a revision of this manuscript.
Thank you.
Please see the attachment
Thank you.
